# CXCR4 Expression as a Prognostic Biomarker in Soft Tissue Sarcomas

**DOI:** 10.3390/diagnostics14111195

**Published:** 2024-06-06

**Authors:** Anna C. Virgili, Juliana Salazar, Alberto Gallardo, Antonio López-Pousa, Raúl Terés, Silvia Bagué, Ruth Orellana, Caterina Fumagalli, Ramon Mangues, Lorena Alba-Castellón, Ugutz Unzueta, Isolda Casanova, Ana Sebio

**Affiliations:** 1Department of Medical Oncology, Hospital de la Santa Creu i Sant Pau, 08041 Barcelona, Spain; avirgili@santpau.cat (A.C.V.); alopezp@santpau.cat (A.L.-P.); rteres@santpau.cat (R.T.); 2Department of Medicine, Faculty of Medicine, Universitat Autònoma de Barcelona, 08035 Barcelona, Spain; 3Translational Medical Oncology Laboratory, Institut de Recerca Sant Pau (IR Sant Pau), 08041 Barcelona, Spain; 4Department of Pathology, Hospital de la Santa Creu i Sant Pau, 08041 Barcelona, Spain; agallardoa@santpau.cat (A.G.); sbaguer@santpau.cat (S.B.); rorellana@santpau.cat (R.O.); cfumagalli@santpau.cat (C.F.); 5Institut de Recerca Sant Pau (IR Sant Pau), 08041 Barcelona, Spain; rmangues@santpau.cat (R.M.); lalba@santpau.cat (L.A.-C.); uunzueta@santpau.cat (U.U.); icasanova@santpau.cat (I.C.); 6CIBER de Bioingeniería, Biomateriales y Nanomedicina (CIBER-BBN), Bellaterra, Cerdanyola del Vallès, 08193 Barcelona, Spain; 7Josep Carreras Leukaemia Research Institute (IJC), 08916 Badalona, Spain

**Keywords:** soft tissue sarcomas, synovial sarcomas, undifferentiated pleomorphic sarcomas, prognostic factor, CXCR4

## Abstract

Poor long-term survival in localized high-risk soft tissue sarcomas (STSs) of the extremities and trunk highlights the need to identify new prognostic factors. CXCR4 is a chemokine receptor involved in tumor progression, angiogenesis, and metastasis. The aim of this study was to evaluate the association between CXCR4 expression in tumor tissue and survival in STSs patients treated with neoadjuvant therapy. CXCR4 expression was retrospectively determined by immunohistochemical analysis in serial specimens including initial biopsies, tumors post-neoadjuvant treatment, and tumors after relapse. We found that a positive cytoplasmatic expression of CXCR4 in tumors after neoadjuvant treatment was a predictor of poor recurrence-free survival (RFS) (*p =* 0.003) and overall survival (*p =* 0.019) in synovial sarcomas. We also found that positive nuclear CXCR4 expression in the initial biopsies was associated with poor RFS (*p =* 0.022) in undifferentiated pleomorphic sarcomas. In conclusion, our study adds to the evidence that CXCR4 expression in tumor tissue is a promising prognostic factor for STSs.

## 1. Introduction

Soft tissue sarcomas (STSs) include more than 80 histological subtypes of rare tumors derived from mesenchymal tissues [1]. Standard treatment for localized disease is wide local excision [2], and known prognostic factors are tumor size, location depth, and high grade. Several validated nomograms are used to predict the risk of recurrence [3]. Neoadjuvant chemotherapy, based on a combination of anthracyclines and ifosfamide and/or radiotherapy are additional treatment options for patients with STSs of the extremities and trunk at high risk of relapse [4]. However, despite multimodality treatment, 40 to 50% of patients develop distant metastases. Current treatment for these patients is systemic chemotherapy but long-term survival remains poor [5]. To improve the survival rates, it is therefore essential to select the best treatment option for each patient based on histological subtype, tumor biology, and prognostic factors.

The CXCR4 chemokine receptor is a promising prognostic biomarker in cancer. CXCR4 is a G-protein coupled receptor located on the cell surface, but it is also expressed in cytoplasm and the nucleus, and its subcellular localization contributes to the regulation of CXCR4-mediated signaling pathways [6,7,8]. CXCL12, also known as chemokine stromal cell-derived factor-1 (SDF-1), is the natural ligand of CXCR4. The CXCR4/CXCL12 complex mediates tumor growth and metastasis [8,9,10,11,12,13,14]. High CXCR4 expression has been associated with poor prognosis in several malignant tumors [15,16,17,18] including mesenchymal neoplasms [19,20,21,22].

Additional research shows that CXCR4 is overexpressed in cancer stem cells in various tumors, driving a more tumorigenic phenotype [23,24,25]. These cells have been identified in many mesenchymal tumors [26,27,28,29]. Cancer stem cells may contribute to treatment resistance and cancer relapse due to their ability to self-renew and switch to a quiescent state [30]. To improve conventional cancer treatments, therapeutic approaches targeting the CXCR4 chemokine receptor such as small molecules, peptidic antagonists, and monoclonal antibodies are currently under development [31]. CXCR4-targeted nanoparticles are also under investigation in several cancer types [32,33,34,35]. Since 2012, our oncogenesis and antitumor drugs research group has been developing a therapeutic nanoparticle for CXCR4-targeted drug delivery. This is currently being tested in several CXCR4-overexpressing cancer types, and preclinical results are encouraging [36,37,38,39,40].

Strong predictive and prognostic factors are lacking in STSs. Several studies that have focused on the identification of molecular biomarkers in STSs have led to the development of transcriptomic signatures [41,42,43]. These signatures are awaiting validation in prospective trials. However, continued investigation into additional and more informative biomarkers is needed. Aware that CXCR4 could be a valuable prognostic factor in cancer [15,16,17,18,19] and a therapeutic target [44,45], it has been hypothesized that CXCR4 expression could be a useful tool to help decision-making in patients with high-risk soft tissue tumors [19,46]. In a well-defined cohort of high-risk STSs patients, we determined CXCR4 nuclear and cytoplasmatic expression in serial samples including initial biopsies, tumors post-neoadjuvant surgical treatment, and tumors after relapse. Based on this comprehensive approach, our objective was to study whether CXCR4 could be useful to detect patients at high risk of relapse or death, and to predict sensitivity to neoadjuvant systemic treatment.

## 2. Materials and Methods

### 2.1. Study Population

Forty-eight patients with STSs of the extremity or trunk and treated with neoadjuvant chemotherapy (anthracyclines and/or ifosfamide) or chemoradiotherapy were recruited at Hospital de la Santa Creu i Sant Pau between January 2006 and March 2021. The archived formalin-fixed paraffin-embedded (FFPE) tumor tissue samples available for the study were 30 initial biopsies, 41 post-neoadjuvant treatment tumors, and 13 post-relapse tumors.

### 2.2. Immunohistochemistry

Tissue microarrays (TMAs) were constructed to analyze the expression of CXCR4 by punching 3 mm diameter cylindrical tissue cores from paraffin blocks and re-embedding them into the microarrays. Two tumor samples were included into each TMA to verify antibody specificity and staining quality. Immunohistochemical staining of the TMAs was then performed in a DAKO Autostainer Link48 (Agilent, Santa Clara, CA, USA) using the anti-CXCR4 antibody (1:200; Abcam, Cambridge, UK, ab124824) following the manufacturer’s instructions. All TMAs were independently examined by two investigators (AG and ACV) under an optical microscope. CXCR4 expression was evaluated both in the cytoplasm (included expression in cell membrane) and in the nucleus of tumor cells. The intensity of the staining and the percentage of positive cells were used to define the immunohistochemical score (H-score). We defined a dichotomous variable considering tumors having an H-score = 0 as “negative expression” and tumors with an H-score ≥ 1 as “positive expression”.

Representative pictures were taken using an Olympus DP73 camera and processed using Olympus cellSens Entry 1.18 software (Olympus, Tokyo, Japan).

### 2.3. Statistics

Recurrence-free survival (RFS) was defined as the date of starting neoadjuvant chemotherapy until the date of local or distant recurrence, whichever occurred first. Overall survival (OS) was calculated from the date of diagnosis (biopsy) to date of death from any cause or last clinical follow-up. We used Kaplan–Meier curves and a log-rank test for OS and RFS analyses. Cox-regression was applied for the multivariate analyses. Survival analyses for age at diagnosis (<60 years versus ≥60 years), sex, Eastern Cooperative Oncology Group (ECOG) performance status (PS) (1–2 versus 0), chemotherapy regimen (anthracyclines plus ifosfamide versus high dose ifosfamide) and neoadjuvant radiotherapy were performed. The statistically significant clinicopathological variables were included as covariables in the multivariate analyses. Statistical significance was set at *p* ≤ 0.05. All statistical analyses were performed using IBM SPSS Statistics (version 26.0).

## 3. Results

### 3.1. Clinical Results

Baseline characteristics of the patients and tumors are shown in Table 1. Median OS was 65.2 (range 38.0–92.4) months, and median RFS was 30.1 (range 21.1–39.1) months. Thirty patients (62.5%) relapsed and 26 died (54.2%) during follow-up. The analyses of the clinicopathological variables that, according to existing knowledge, would influence survival in STSs patients treated with neoadjuvant chemotherapy showed that ECOG PS and age were significantly associated with OS. Patients with an ECOG PS of 1–2 showed worse survival than patients who had ECOG PS 0 [39.4 months (95% CI: 10.5–68.2) versus not reached; *p =* 0.035]. Patients under 60 years of age had a median OS of 113 [95% CI: not applicable (NA)] months, whereas patients aged 60 or over had a median OS of 24.6 (95% CI: 0.0–50.7) months (*p =* 0.004). These variants were included in the multivariate analyses for OS. As none of the clinicopathological variables analyzed showed statistically significant associations with RFS, multivariate analyses for RFS were not performed.

### 3.2. CXCR4 Expression and Localization

Figure 1 shows CXCR4 expression in some histological subtypes of STSs.

Eighty percent (24/30) of initial biopsies had positive CXCR4 expression (11 nuclear, 6 cytoplasmatic, and 7 both nuclear and cytoplasmatic). In surgical samples after neoadjuvant treatment, CXCR4 positivity was 49% (20/41) (5 nuclear, 14 cytoplasmatic, and 1 both nuclear and cytoplasmatic). Fifty-four percent (7/13) of tumors after relapse were positive for CXCR4 (2 nuclear and 5 cytoplasmatic) (Figure 2). Our data showed that the predominant localization pattern of CXCR4 expression in the initial biopsies was nuclear, whereas in the post-neoadjuvant treatment tumors, it was cytoplasmatic. These differences, however, were not statistically significant. No specific pattern of CXCR4 expression was detected among the various histological subtypes of STSs or according to the treatment received.

### 3.3. CXCR4 in the Cohort

Numerical differences were found in RFS and OS according to CXCR4 cytoplasmatic expression in post-neoadjuvant treatment tumors, but they did not reach statistical significance. Five-year RFS was 20% for patients with positive CXCR4 cytoplasmatic expression compared to 46% for patients with negative cytoplasmatic expression (*p =* 0.1). The 5-year OS was 30% for patients with cytoplasmatic CXCR4-positive expression compared to 63% for patients with CXCR4-negative expression (*p =* 0.06) (Figure 3).

When analyzing tumors at relapse, nuclear CXCR4 expression predicted shorter OS, although the number of samples was scarce. During the follow-up period, both patients with CXCR4-positive expression (2/2) and 70% (7/10) of patients with CXCR4-negative expression died [18 months (95% CI: NA) in positive CXCR4 versus 53.9 months (95% CI: 17.12–90.68) in negative CXCR4; *p =* 0.017]. Inversely, positive cytoplasmatic CXCR4 expression was associated with higher OS. Sixty percent (3/5) of patients with CXCR4-positive expression, and 88% (7/8) with CXCR4-negative expression died [113 months (95% CI: 27.95–198.02) in positive CXCR4 versus 39.4 months (95% CI: 29.20–49.56) in negative CXCR4; *p =* 0.039].

In the multivariate analyses, none of the associations observed with OS retained statistical significance after adjusting for the clinicopathological covariates.

### 3.4. CXCR4 in Synovial Sarcomas

Patients with synovial sarcomas (*N* = 9) that expressed cytoplasmatic CXCR4 in surgical biopsies after neoadjuvant treatment showed unfavorable RFS [17.3 months (95% CI: 7.22–27.40) in positive CXCR4 versus not reached in negative CXCR4; *p =* 0.003] (Figure 4A). At 5 years, all patients with cytoplasmatic CXCR4-positive expression relapsed compared to a RFS rate of 80% for patients with CXCR4-negative expression. The same association was observed for OS; positive cytoplasmatic CXCR4 expression predicted shorter survival [42.6 months (95% CI: 31.47–53.64) in positive CXCR4 versus not reached in negative CXCR4; *p =* 0.019] (Figure 4B). At 5 years follow-up, all CXCR4-positive patients and 75% of CXCR4-negative patients died.

No statistically significant associations were observed between nuclear CXCR4 expression after neoadjuvant chemotherapy and survival.

### 3.5. CXCR4 in Undifferentiated Pleomorphic Sarcomas (UPS)

Statistically significant differences in RFS were found according to the nuclear CXCR4 expression in initial biopsies in patients with UPS (*N* = 12). Patients whose tumors expressed CXCR4 in the cell nucleus at diagnosis showed worse RFS than patients whose tumors did not express the protein [9.2 months (95% CI: 4.40–14.10) versus 20.1 months (95% CI: 5.51–34.62), respectively; *p =* 0.022] (Figure 5). This association was not observed in OS (*p =* 0.3).

No statistically significant associations were found between cytoplasmatic CXCR4 expression at diagnosis and survival.

## 4. Discussion

In this study, we found that CXCR4 expression predicted poor prognosis in several histological subtypes of STSs. In synovial sarcomas in the post-neoadjuvant setting, we observed that cytoplasmic CXCR4 was associated with survival. Additionally, nuclear CXCR4 expression was associated with RFS in the initial biopsies in UPS.

Previous studies have shown that CXCR4 correlates with the expression of the angiogenic factor VEGF in STSs [22,47], and that the CXCR4/CXCL12 complex may participate in tumor dissemination and metastasis in rhabdomyosarcoma cell lines [10,12]. In synovial sarcoma, cancer stem cells expressing CXCR4 have been shown to have tumor initiation and self-renewal capacity [29]. Oda et al. [22] reported that CXCR4 expression was associated with tumor site and tumor necrosis, and that CXCR4 overexpression and high stage could be prognostic factors in malignant non-round cell tumors. These results were in line with associations described between high CXCR4 expression and poor outcome in patients with rhabdomyosarcoma [21] and synovial sarcoma [29,48], and in a 12-study meta-analysis of bone sarcomas and STSs [19].

Our results are consistent with these studies and show that cytoplasmatic CXCR4 expression after neoadjuvant treatment may be a useful biomarker to detect high-risk patients. We suggest that CXCR4 expression in the post-neoadjuvant setting could identify tumors with cancer stem cells that are resistant to neoadjuvant treatment and may be involved in tumor relapse [49], eventually impacting survival. These observations are especially relevant in synovial sarcoma characterized by a high incidence of metastatic disease [50], and a low 5-year OS rate [3,51]. We also found an association between positive nuclear CXCR4 expression in the initial biopsy and worse RFS in patients with UPS. Thus, synovial sarcoma and UPS patients whose tumors express CXCR4 may benefit the most from the development of CXCR4-targeted therapies and/or intensive surveillance.

Another finding of interest in our study was the observation that in tumors after relapse, nuclear CXCR4 expression was associated with worse OS, while cytoplasmatic CXCR4 expression was associated with better OS. CXCR4 transfers from the cytoplasm to the nucleus after CXCL12 binding [7,52,53], and once in the nucleus, it facilitates the activation of the hypoxia-inducible factor-1 (HIF-1) pathway [54]. Additionally, CXCR4 nuclear staining has been detected in cells of metastatic tumors [6,53]. Based on our results, we can speculate that the nuclear translocation of CXCR4 may also be relevant in the metastatic disease in STSs, and may provide additional treatment options in this late stage of the disease.

Most studies show that nuclear, cytoplasmic, or membrane CXCR4 expression correlates with poor survival, and that CXCR4 location differs by cancer type [55]. Interestingly, the diverse CXCR4 location may activate different signaling pathways, cellular responses, and cancer progression [55]. Additionally, subcellular location determines the targeted therapy approach used to ensure that it reaches its target, which would be more difficult for nuclear location [56]. Our observation of associations between CXCR4 nuclear or cytoplasmic expression and poor prognosis in STSs differs from those described between CXCR4 membrane expression and poor prognosis in epithelial cancers [55,57]. Nonetheless, these results are supported by the cellular localization of CXCR4, differing according to cell type, tumor histology, and tumor stage. Sarcomas derive from non-transformed mesenchymal stem cells [58] that express CXCR4 in the nucleus [59]. Normal epithelial cells or tissues and epithelial-derived primary tumors express CXCR4 in the membrane [60]. Metastatic carcinomas, which have undergone epithelial–mesenchymal transition, express CXCR4 in the nucleus [6,52,61].

Here, we provide evidence of the prognostic value of CXCR4 expression in tumors of patients with STSs who received neoadjuvant treatment. However, our study has some limitations. First, it involved a small number of patients due to the limited availability of FFPE for research purposes, but as far as we know, this is the first study to determine nuclear and cytoplasmatic expression of CXCR4 in serial samples. This approach allowed us to detect differences for some histological subtypes, showing its suitability in highly heterogeneous tumors such as STSs. Second, the number of patients for each histological subtype was low due to the rarity of these tumors. Nevertheless, our results provide a basis for future therapeutic strategies such as the development of CXCR4 inhibitors or CXCR4-targeted therapies and their prospective validation in clinical trials.

## 5. Conclusions

Our findings strengthen the importance of CXCR4 in STSs as a prognostic factor in some histological subtypes such as synovial sarcomas and UPS, suggesting a new line of research toward a therapeutic approach.

## Figures and Tables

**Figure 1 diagnostics-14-01195-f001:**
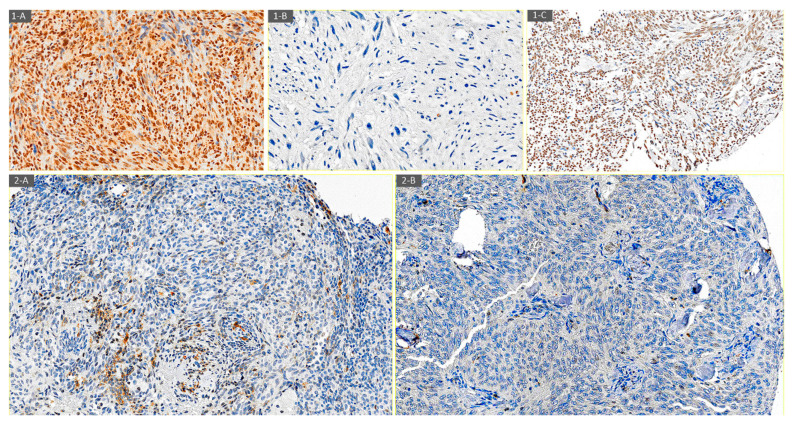
Examples of CXCR4 expression determined by immunohistochemistry. (**1-A**) CXCR4 nuclear and cytoplasmatic positive immunostaining in high-grade spindle cell sarcoma (20×). (**1-B**) CXCR4 negative immunostaining in undifferentiated pleomorphic sarcoma (20×). (**1-C**) CXCR4 nuclear positive immunostaining in synovial sarcoma (20×). (**2-A**) CXCR4 cytoplasmatic positive immunostaining in the initial biopsy of a 46-year-old woman with biphasic synovial sarcoma (20×). (**2-B**) CXCR4 negative immunostaining in post-neoadjuvant treatment biopsy of a 46-year-old woman with biphasic synovial sarcoma (20×).

**Figure 2 diagnostics-14-01195-f002:**
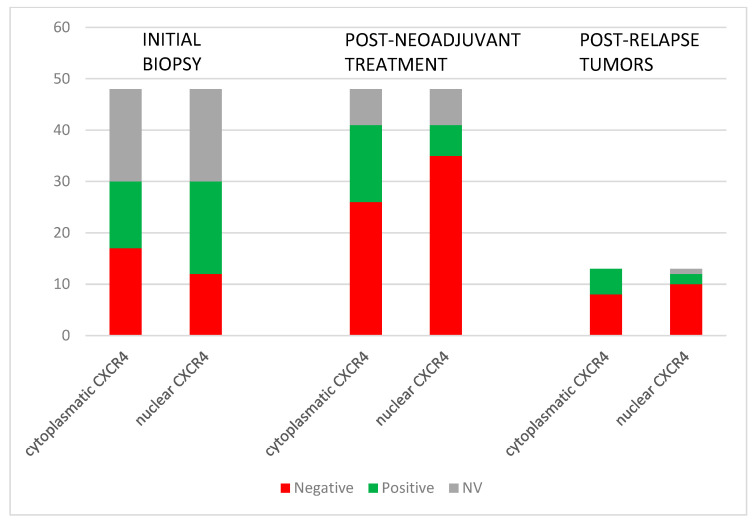
CXCR4 expression in the evolution of the STSs tumors: before and after neoadjuvant treatment and post-relapse.

**Figure 3 diagnostics-14-01195-f003:**
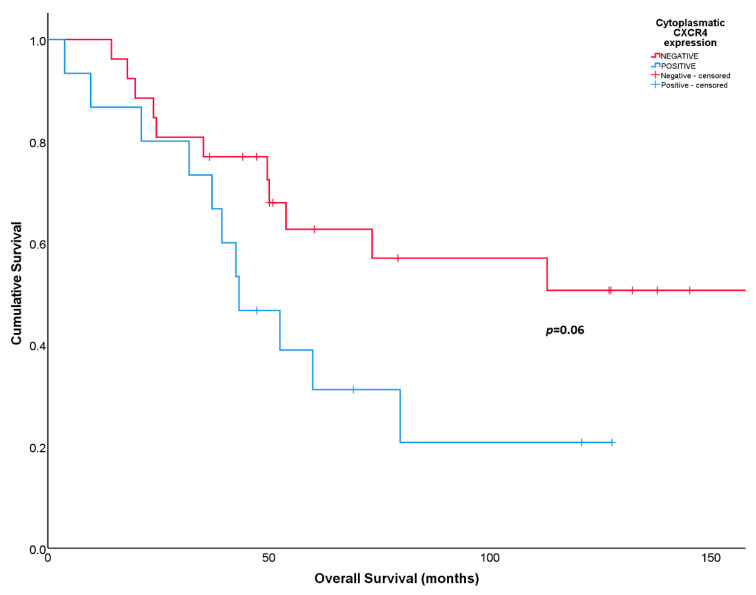
Overall survival according to cytoplasmatic CXCR4 expression in post-neoadjuvant surgical samples.

**Figure 4 diagnostics-14-01195-f004:**
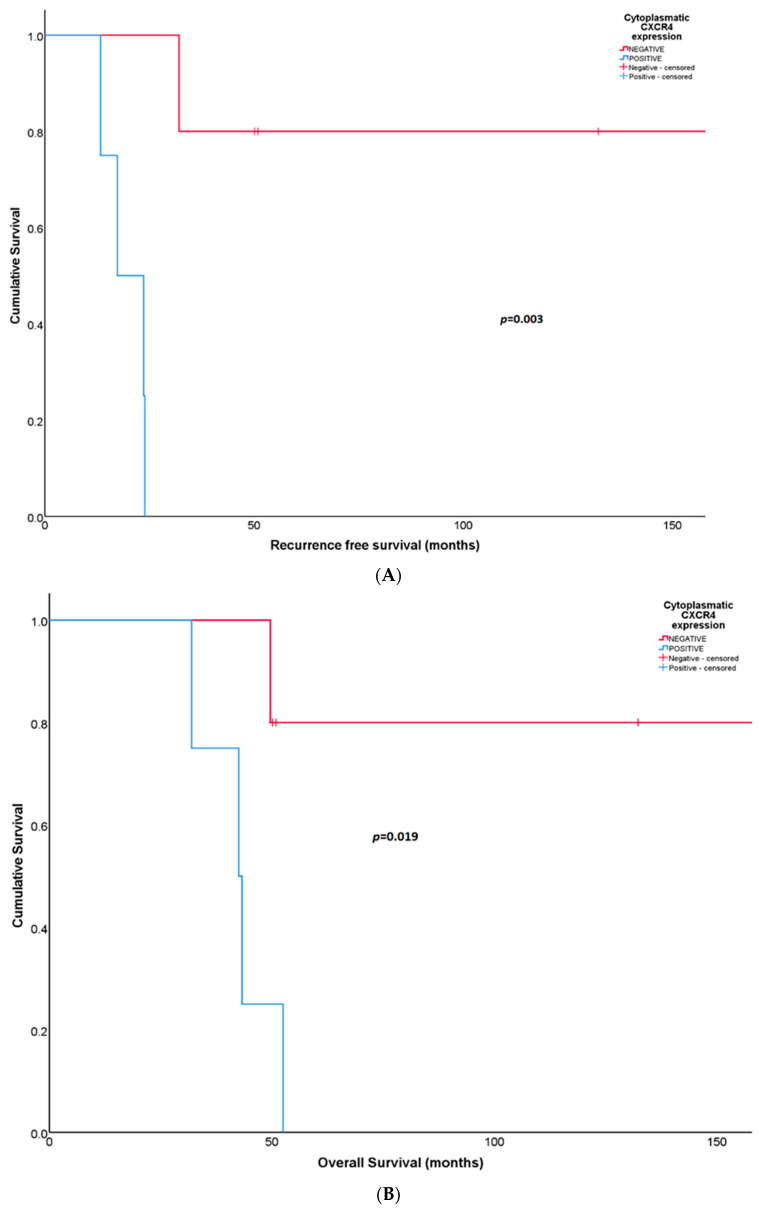
(**A**) Recurrence-free survival according to cytoplasmatic CXCR4 expression in post-neoadjuvant surgical samples from patients with synovial sarcoma. (**B**) Overall survival according to cytoplasmatic CXCR4 expression in post-neoadjuvant surgical samples obtained from patients with synovial sarcoma.

**Figure 5 diagnostics-14-01195-f005:**
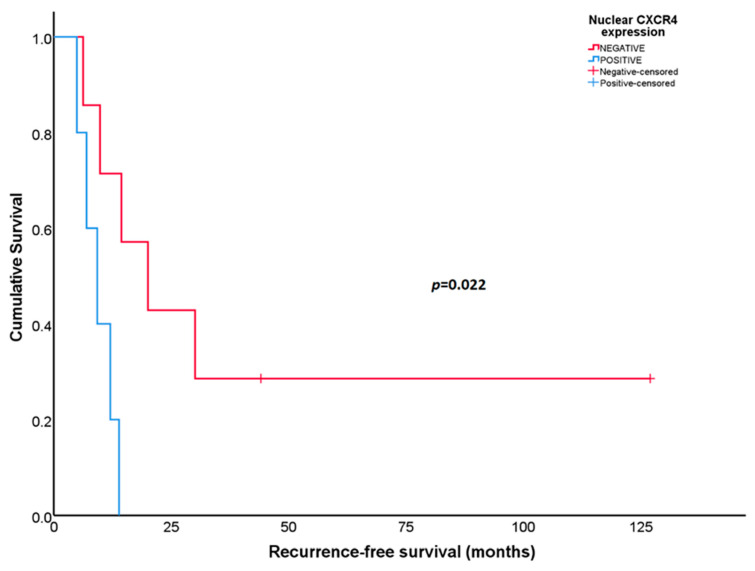
Recurrence-free survival according to nuclear CXCR4 expression in the initial biopsies for patients with undifferentiated pleomorphic sarcoma.

**Table 1 diagnostics-14-01195-t001:** Demographic and clinical characteristics of high-risk patients with soft tissue sarcoma.

Patient and Tumor Characteristics	*N* = 48	%
**Age (years)**		
Median	53	
Range	20–77	
<60	34	70.8
≥60	14	29.2
**Sex**		
Male	30	62.5
Female	18	37.5
**ECOG * Performance status**		
0	15	31.3
1	17	35.4
2	1	2.0
Unknown	15	31.3
**Histology**		
Undifferentiated pleomorphic sarcoma	17	35.4
Synovial sarcoma	10	20.8
Spindle cell sarcoma, NOS **	9	18.8
Others	12	25.0
**Chemotherapy**		
Epirubicin-ifosfamide	25	52.1
High-dose ifosfamide	15	31.2
Others	8	16.7

* ECOG—Eastern Cooperative Oncology Group; ** NOS—Not-otherwise specified.

## Data Availability

The data presented in this study are not publicly available due to ethical committee regulations but are available on request from the corresponding authors on reasonable request.

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
