# Peer review of "CXCR4 Expression as a Prognostic Biomarker in Soft Tissue Sarcomas"

_diagnostics, 2024, doi:10.3390/diagnostics14111195_

Round 1

Reviewer 1 Report

Comments and Suggestions for Authors

The manuscript titled” CXCR4 expression as a prognostic biomarker in soft tissue sarcomas” by Sebio, evaluates CXCR4 expression as a prognostic factor in soft tissue sarcomas (STS) treated with neoadjuvant therapy. It finds that positive cytoplasmic CXCR4 expression after treatment predicts poor recurrence-free survival(RFS) and overall survival in synovial sarcomas, while nuclear CXCR4 expression in initial biopsies predicts poor RFS in undifferentiated pleomorphic sarcomas. CXCR4 is highlighted as a promising prognostic biomarker for STS. This is a novel aspect in a field of intense research. Still some issues need to be clarified, as listed below, before the manuscript can be accepted for publication in diagnostics.

1.      What other prognostic biomarkers are there for STS? How do these biomarkers compare to CXCR4? Why was CXCR4 chosen as the subject of this study?

2.      For method part, do these patients have any other diseases? Are they receiving any other treatments concurrently? Could the selection of patients impact your research findings?

3.      For CXCR4 Expression Levels Results, how did the levels of CXCR4 expression vary among different subtypes of soft tissue sarcomas?

4.      Are there any other treatments for STS? Do different treatment options affect the expression of CXRC4 in STS?  Can this study be applied in the future to optimize the selection of treatment strategies for STS?

Reviewer 2 Report

Comments and Suggestions for Authors

The authors studied the relationship between CXCR4 expression and RFS/OS in patients with STS. Microarray analysis was performed to determine the presence of CXCR4 in the cytosol and nucleus. Statistical tests were conducted to determine the correlation between CXCR4 expression and prognostic observations. The concept is significant and could be of interest to a wide range of audiences. However, additional revisions may be needed to make the manuscript more thorough and scientifically sound.

1.      It was stated in line 64 that 'Aware that strong predictive and prognostic factors are lacking in STS…'. However, previous studies, such as [cite example: https://doi.org/10.1155/2011/593708], have explored the relationship between CXCR4 and STS, which wasn't acknowledged. In fact, there is a body of research focused on CXCR4 and STS, suggesting that the findings presented in the manuscript may not be entirely novel. Therefore, it's imperative to discuss this background information more thoroughly in the manuscript.

2.      It would be beneficial to complement the images shown in Fig 1 with a stacked bar plot representing the data discussed in section 3.2. This additional visualization can offer a more comprehensive understanding of the results and aid in the interpretation of the findings. Same for the other sections.

3.      It was mentioned in line 218 that 'cytoplasmic CXCR4 expression after neoadjuvant treatment could predict survival in patients with STS,' which may not be entirely accurate. While the study explored the association between cytoplasmic CXCR4 expression and survival outcomes in STS patients, predictive classification or regression models focused on CXCR4 and RFS/OS were not developed or evaluated. Metrics such as prediction accuracy, recall, sensitivity, etc., were not established. Therefore, the conclusion regarding the predictive capability of cytoplasmic CXCR4 expression requires refinement.

4.      It was noted that clinically significant covariates were included in the statistical testing; however, this aspect was not further discussed in the manuscript. To provide a comprehensive analysis, additional discussion regarding the selection and inclusion of these covariates is warranted. Explaining how these covariates were chosen, their potential impact on the results, and any adjustments made for them in the statistical models can enhance the readers' understanding of the analysis and interpretation of the findings.

Round 2

Reviewer 2 Report

Comments and Suggestions for Authors

The authors have addressed all my concerns.